# Simulating DNA Chip Design Using All-Electronic Graphene-Based Substrates

**DOI:** 10.3390/molecules24050951

**Published:** 2019-03-08

**Authors:** Ernane de Freitas Martins, Gustavo Troiano Feliciano, Ralph Hendrik Scheicher, Alexandre Reily Rocha

**Affiliations:** 1Institute of Theoretical Physics, São Paulo State University (UNESP), Campus São Paulo, 01140-070 São Paulo, Brazil; ernanefmg@gmail.com; 2Division of Materials Theory, Department of Physics and Astronomy, Uppsala University, SE-751 20 Uppsala, Sweden; ralph.scheicher@physics.uu.se; 3Institute of Chemistry, São Paulo State University (UNESP), Campus Araraquara, 14800-060 Araraquara, Brazil; gtroiano@iq.unesp.br

**Keywords:** DNA chip, graphene, QM/MM, non-equilibrium Green’s functions

## Abstract

In this paper, we present a theoretical investigation of an all-electronic biochip based on graphene to detect DNA including a full dynamical treatment for the environment. Our proposed device design is based on the changes in the electronic transport properties of graphene interacting with DNA strands under the effect of the solvent. To investigate these systems, we applied a hybrid methodology, combining quantum and classical mechanics (QM/MM) coupled to non-equilibrium Green’s functions, allowing for the calculations of electronic transport. Our results show that the proposed device has high sensitivity towards the presence of DNA, and, combined with the presence of a specific DNA probe in the form of a single-strand, it presents good selectivity towards specific nucleotide sequences.

## 1. Introduction

DNA stores all the genetic information of a living organism [1], and identifying the four different nucleotides that combine into base pairs and form the double-stranded helical structure is a required aspect for introducing so-called personalized medicine: the individualized and precise diagnosis of numerous illnesses [2], and corresponding customized treatments [3,4]. In principle, fully sequencing of the whole genome would provide all the information required for such an endeavor. Nonetheless, that is still a cumbersome, slow and expensive process [5,6,7,8], even though there are proposals for next-generation single-shot sequencing [9,10,11].

Furthermore, most known genetic anomalies are confined to a small proportion of the DNA, in particular small changes—such as mutations—to particular segments of a gene [2,12]. Thus, identifying mutations in the genome can be achieved by a so-called biochip. In these types of devices, the detection is based on a process of hybridization between a specifically designed single-stranded DNA (ssDNA) probe and a strand originating from the sample one wishes to measure [13]. Detection then occurs by differentiating between—usually via an optical measurement—the isolated strands and the double-stranded DNA (dsDNA) that is formed if they are complementary in an all-electronic device. Such molecular recognition is obtained by measuring a difference in conductance between the two configurations as they interact with a conducting substrate [13].

A key issue for proposing new devices for health care is the choice of the material to be used. A very promising candidate is graphene—a hexagonal arrangement of carbon atoms discovered in 2004 by Novoselov et al. [14]. Graphene has the advantage of having the smallest possible thickness, which allows it to interact directly with the sample (DNA) due to its high available superficial area. Furthermore, graphene has high electronic mobility [15,16], which makes it a promising candidate for all-electronic devices for applications in biological systems [17,18,19,20,21,22,23], such as DNA sequencing [24,25,26,27,28].

The main problem in simulating such systems is the fact that one requires a calculation of the transport properties of electrons flowing along a graphene sheet in the presence of a large biomolecule—in our case, DNA. Most importantly, we must consider the solvent, which is ubiquitous in such devices, and can have significant structural and electronic effects. A possible way—and the one used in this work—is to combine quantum mechanics/molecular mechanics (QM/MM) [29,30] and the non-equilibrium Green’s functions (NEGF) [31,32,33,34,35] formalism allowing for the explicit inclusion of solvent effects in electronic transport calculations.

In this work, we employed such a methodology in the study of two possible biochip designs for the detection of DNA strands, as shown in Figure 1. Both are based on the idea of molecular recognition using the hybridization of DNA. It is worth mentioning that in our proposal the device is designed to detect known sequences of DNA—based on hybridization processes—being different from the devices proposed for DNA sequencing. In the first one, a ssDNA segment acting as the probe is anchored on a conducting substrate. When it interacts with a complementary strand it forms (see Figure 1a) a double strand [36]. In the second approach, the sample is mixed with a solution containing the suspended ssDNA probe. The single or double strands—if hybridization occurs—are then allowed to interact with the device, as shown in Figure 1b. In both cases, the sensing mechanism is based on measuring the difference in electronic conductance depending on whether the ssDNA or the dsDNA couples to the substrate. Here, graphene is used as our conducting substrate. Using a combination of QM/MM + NEGF, we are able to demonstrate a high sensitivity and selectivity for these devices proving the way for all-electronic nanoscale single-molecule biochips.

## 2. Methodology

The model biochip setup used in this work was composed by a graphene sheet with dimensions 31.98×46.86 Å connected to source and drain potentials, which here were taken as pristine semi-infinite graphene sheets. The size of the simulation box was chosen as to strike a balance between achieving a description of a as-realist-as-possible system, and computational cost, as our density functional theory (DFT) calculations required configuration-sampled simulations containing more than 800 quantum mechanically described atoms. An ssDNA (or dsDNA) was placed above the graphene sheet and the box was filled with water and NaCl (0.2 M) to simulate physiological conditions. We introduced an unbalance between the positive and negative ions in order to equilibrate the negative charges of the DNA phosphate groups. As a proof of concept, we used poly-guanine for the ssDNA and the corresponding poly-GC for the dsDNA case, in the vertical setup. For the horizontal setup, we used poly-adenine for the ssDNA and the corresponding poly-AT for the dsDNA, as the proposed setup should be able to detect different sequences. Figure 2a shows the scheme for construction of the simulated device using the ssDNA in the vertical setup as example.

Given the dynamic nature of the problem, one requires sampling over a set of structural configurations, here obtained by Molecular Dynamics (MD) calculations, which is a technique accurate enough to sample the configurational space in biological systems [17]. In the MD calculations, the graphene sheet is kept fixed during the simulation, so the box vectors in the directions of the plane of graphene are also fixed. We apply harmonic restraints (*k* = 10 kJ mol${}^{-1}$ nm${}^{-2}$) in the DNA coordinates for z direction (electronic transport direction) to prevent structures in which the DNA is localized above the electrodes (identical electrodes for all structures are required).

We initially performed a 10 ns NPT (Parrinello–Rahman barostat and Noose–Hoover thermostat) simulation to equilibrate the system’s pressure, followed by a 100 ns NVT production run (same thermostat and restrictions) to generate the structures. The NVT simulation was performed using as a starting point a structure extracted from the NPT simulation in which the length of the box in the perpendicular direction to the graphene sheet (the only direction allowed to change) was equal to the average length. For all simulations, we used the AMBER99SB [37] force field and SPC [38,39] water model, and periodic boundary conditions (PBC). The periodic boundary conditions ensured that we were simulating a true graphene sheet, and there were no edges to the graphene sheet. It also meant that there was no problem if the DNA segment overshot the simulation box.

We extracted 50 snapshots from the MD production run for each of the four setups to obtain its electronic structure. Then, we applied a QM/MM approach where the environment is described by classical force field partial charges acting electrostatically over a subset of the system described by quantum mechanics, where electrons are explicitly taken into account [30,40]. The quantum-mechanical region of our system was treated by first-principles calculations based on DFT [41,42], and the MM electrostatic potential was evaluated from the force field’s partial charges, using Coulomb’s law as implemented in Siesta [29,43,44]. In our DFT calculations, the calculated potential was directly added to the Hartree potential, until self-consistency was achieved in the electronic charge density of the QM region. We used the vdW DRSLL [45] functional in our DFT calculations with DZ basis set for graphene and DZP for the remaining atoms (DNA and Na atoms), using Troullier–Martins norm-conserving [46] pseudo potentials. The QM/MM partition chosen here (see Figure 2b) kept all solvent atoms on the MM region, whereas the graphene sheet and the whole DNA strands were treated quantum mechanically (and enough counter ions—the ones closest to the DNA—to neutralize the backbone charges). The electronic transport properties of the systems were then obtained using the Green’s function formalism to calculate the low-bias conductance [32,33,35,47]. In this approach, one associates the conductance to the probability that the wave function of an electron injected from a metallic defectless source electrode will be transmitted through a potential barrier located in a so-called scattering region and into a drain electrode. Thus, the conductance will be given, at a certain gate voltage $V_{g}$, by (1)$$G\left(V_{g}\right)=\frac{2e^{2}}{h}\sum_{k_{y}}T\left(E+V_{g},k_{y}\right)=\frac{2e^{2}}{h}T\left(E+V_{g}\right)\;,$$where $G_{0}=2e^{2}/h$ is the quantum of conductance, and $T\left(E,k_{y}\right)$ is the total transmission probability at a certain transversal k-point $k_{y}$. We considered periodic boundary conditions in the transverse direction for the electronic transport calculations, where we used 16 k-points along $k_{y}$ and $k_{x}=0$ (normal to the plane of graphene).

Thus, in the electronic transport calculation, our system was divided into three regions: a central scattering region and two semi-infinite electrodes. In our case, the electrodes were taken as pristine graphene for the right- and left-hand sides (see Figure 3c). The scattering region consisted of the graphene sheet and the DNA molecule (indirectly including the effect of the solvent via the external potential). The transmission could then be calculated via the Green’s functions for this open system [31,48,49,50], (2)$$T\left(E,k_{y}\right)=\Gamma_{L}\left(E,k_{y}\right)G^{A}\left(E,k_{y}\right)\Gamma_{R}\left(E,k_{y}\right)G^{R}\left(E,k_{y}\right)\;,$$where $\Gamma_{\alpha }=i\left[\Sigma_{\alpha }-\Sigma_{\alpha }^{†}\right]$ ($\alpha \equiv \left\{L,R\right\}$), $\Sigma_{L/R}$ are the self-energies [33,34,35]—the effect of the semi-infinite electrodes on the scattering region—and, (3)$$G^{R}\left(E,k_{y}\right)={\left[\epsilon^{+}S_{S}\left(k_{y}\right)-H_{S}\left(k_{y}\right)-\Sigma_{L}\left(E,k_{y}\right)-\Sigma_{R}\left(E,k_{y}\right)\right]}^{-1}\;,$$ is the retarded Green’s function for the scattering region, and $\epsilon^{+}=E+i\eta$. Finally, the QM Hamiltonian $H_{S}$ was obtained by the previous DFT calculations for the different setups, and it included the effects of the solvent.

In summary, we performed a four-step procedure, with the first one being a classical MD calculation to sample the configurations of the system. We then divided the system into a QM and a MM region and performed a single-point DFT calculation including the classical potential calculated for the MM region. After that, we calculated the transmission probability as a function of energy in the resulting Kohn–Sham Hamiltonian. In the final step, we averaged the results to obtain statistically representative data that included the dynamic effects of the solvent.

## 3. Results

To simulate the two setups shown in Figure 1, we initially placed the strands either vertically or horizontally close to the graphene sheet. Side and top views of the two used setups (vertical and horizontal) for ssDNA and dsDNA are shown in Figure 3. For the situation where the ssDNA was anchored on graphene, we used a strand containing six guanine nucleotides. The first nucleotide was used as the anchor and the second as a buffer to prevent π-πinteractions between the complementary strand and graphene that interacts via π stacking with graphene. Thus, the complementary strand forming the double helix only had four cytosine nucleotides. For the horizontal case, we used as ssDNA and dsDNA segments of the same length containing four adenine and four thymine. Figure 3 shows for each simulated system (ssDNA and dsDNA in both vertical and horizontal setups): (a) the isolated DNA sequence; (b) a side view of the simulation box for a typical snapshot; and (c) a top view of the DNA systems above graphene hiding the solvent components for clarity.

In the case of vertical alignment, the ssDNA immediately adsorbed on graphene via π-πstacking during the thermalization procedure and it remained so throughout the production run (Figure 3). We point out that this interaction between graphene and DNA nucleotides is extremely strong, and hard to break. In fact, it is one of the main sources of clogging in proposed graphene-nanopore-based DNA sequencing devices [51]. Subsequent calculations using a double strand presented a similar overall structure, with only a single nucleotide adsorbed on graphene, as shown in the same figure.

On the other hand, during the first MD simulation (NPT ensemble) for ssDNA in the horizontal setup (see Figure 3), we observed the adsorption (via π-πstacking) of three nucleotides on graphene (the fourth nucleotide did not directly interact with the sheet). For steric gain, once the stacking was formed during thermalization, they remained throughout the production run. For dsDNA, however, the hydrogen bonds between the two strands forming the helix prevented π-πinteractions between the nucleotides and graphene. In that case, the DNA continued to interacting with graphene due to the charged groups of DNA, which were placed close to the sheet.

From the MM-obtained structures, we performed static QM/MM-NEGF calculations of the electronic transport. In Figure 4a, we present the total transmission coefficients per unit width as a function of energy for pristine graphene, and the average—over 50 frames—for graphene in solution. An analysis of convergence for these calculations is presented in the Appendix A. Acting as both the device and the electrodes in this case, pristine graphene presented the highest conductance. Thus, when graphene was immersed in solution (water and 0.2 M of NaCl), one could observe that the transmission decreased, particularly for negative gate voltages. This effect can be understood in terms of a local gating that drives down the Dirac cone of graphene inside the scattering region compared to the electrodes (the cone at $E_{F}$) [18].

In Figure 4b,c, the average transmission for all setups are presented. Firstly, we note that, in this figure, the transmission coefficients for the DNA strands are different from the transmission of the system in water, especially for energies below the Fermi level. As it can be observed in Figure 4b, the signals for vertical ssDNA and dsDNA were only slightly different. The reason for this is the fact that the charged groups of the nucleotides were far from graphene, and the presence of water quickly screened the charge. Therefore, when the ssDNA hybridized forming the double strand, the extra charges that came from the added nucleotides did not alter the electronic properties of graphene. In fact, the presence of either form of DNA hindered the chemical gating effect from the solvent, and graphene recovered, to some extent, its characteristic Dirac cone.

The results for horizontal setup, shown in Figure 4c, presented different averages for all energies, indicating that this is most likely the setup in which the detection can be achieved. The horizontal setup presented better detection because the charged groups of DNA were closer to the substrate, which would change the electronic environment of graphene. Most importantly, we point out that the type of interactions between the two systems (ssDNA versus dsDNA) is different.

To quantify the possibility of DNA sensing, the sensitivity (*S*) and selectivity (*s*) of the proposed devices was calculated. We defined the sensitivity *S* as the ability to determine the presence of DNA compared to a basal reference (taken here as the average transmission calculated for graphene in solution (L)), (4)$$S\;\left(E\right)=\left|\frac{{\bar{T}}_{x}\left(E\right)-{\bar{T}}_{L}\left(E\right)}{{\bar{T}}_{L}\left(E\right)}\right|\times 100\;\%\;,$$where ${\bar{T}}_{x}\left(E\right)$ is the average transmission as a function of energy for x={ssDNA,dsDNA} and ${\bar{T}}_{L}\left(E\right)$ is the average transmission for the reference (liquid case, i.e., with no nucleotide). The most important quantity, however, is the selectivity *s*, i.e., determining how distinguishable the signals between dsDNA and ssDNA are:(5)$$s\;\left(E\right)=\left|\frac{{\bar{T}}_{dsDNA}\left(E\right)-{\bar{T}}_{ssDNA}\left(E\right)}{{\bar{T}}_{ssDNA}\left(E\right)}\right|\times 100\;\%\;,$$where ${\bar{T}}_{ssDNA}\left(E\right)$ is the average transmission for ssDNA and ${\bar{T}}_{dsDNA}\left(E\right)$ for dsDNA that in this case is taken as our reference.

It is the selectivity that indicates whether hybridization has occurred and if the sample contains a specific strand of DNA—a mutated gene, for instance.

As can be seen in Figure 5, which presents the selectivity and sensitivity for both setups, the highest sensitivity was achieved for $V_{g}=E-E_{F}\approx -0.31$ eV, ranging from ≈45% to 70%. At the Fermi Level, $E-E_{F}=0.00$ eV, the sensitivity was close to zero for both setups because the transmission aws close to zero. This was expected because of the Dirac cone in the electronic structure of graphene. The observed noise in the calculated selectivity *s* for the vertical setup occurred due to the larger structural fluctuation observed for ssDNA and dsDNA in the vertical setup (see Appendix A for more details).

The calculated selectivity presented in Figure 5 indicates that it is typically smaller compared to S(E). For the vertical setup, in fact, it would be hard to differentiate between the two. Nonetheless, for the horizontal setup, the dsDNA signal was about 20% larger than the one for ssDNA at $V_{g}=E-E_{F}\approx \pm 0.31$ eV. For the Fermi level, the two signals were the same, as expected, presenting selectivity close to zero. These results for selectivity show that the signals for ssDNA and dsDNA were distinguishable at these gate voltages, especially for $E-E_{F}\approx -0.31$ eV, where we had good sensitivity combined with good selectivity. In fact, from the experimental point of view, the horizontal setup, besides presenting the best sensitivity, would also be the most likely setup, as the typical binding energy between graphene and different nucleobases is ∼0.5 eV [52]. This means, once any base starts to interact with graphene, it will likely stick, and would require a large force to draw them apart [51].

Thus, in essence, a two-setup device whereby single-strands of DNA of a sample are allowed to interact, and possibly hybridize with a probe in solution can then be subsequently deposited on a graphene substrate that doubles as an electronic sensor.

## 4. Conclusions

In summary, we present a theoretical study for a DNA sensing device based on graphene fully accounting for the effect of the environment using a hybrid methodology that combines quantum and classical mechanics (the QM/MM method) coupled to the NEGF formalism to simulate electronic transport properties. Our results show that devices based on graphene are capable of detecting ssDNA and dsDNA and to distinguish their signals, presenting good sensitivity and selectivity for a wide range of energy. Those differences for the ssDNA and dsDNA signals are due to the presence of new charged groups close to the graphene sheet when we add a second DNA strand forming a helix in the case of a setup where DNA lies horizontally on the sheet. As those charged groups are mostly located on the backbone, they are largely base independent, and as such our results should remain valid for different base combinations, and should enhance as the length of the strands increase. Finally, it might be possible to improve selectivity and sensitivity by further reducing the dimensionality using, for example, topological line defects in graphene [24].

## Figures and Tables

**Figure 1 molecules-24-00951-f001:**
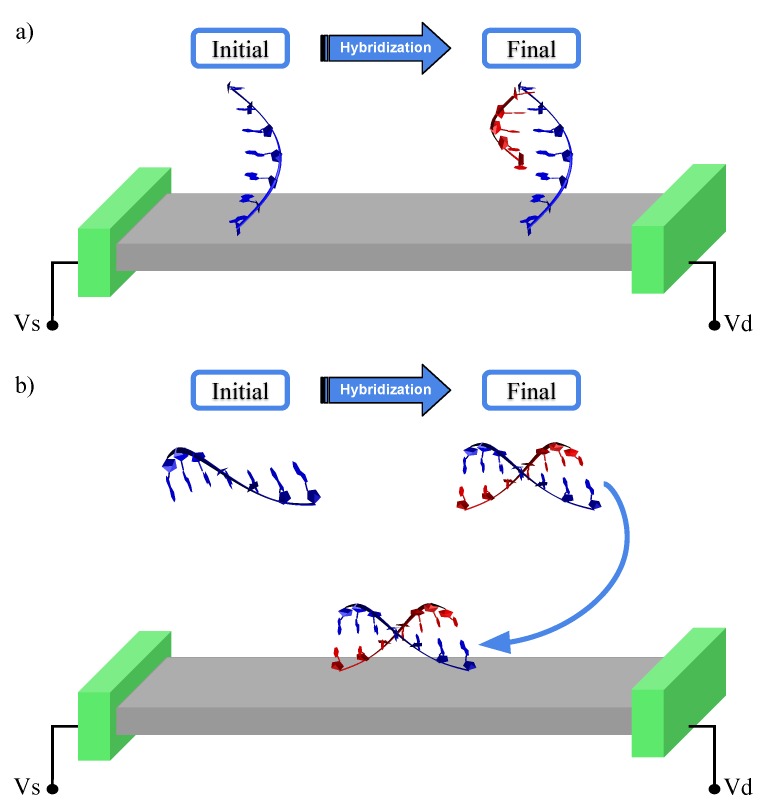
Sketch of all-electronic biochips. An active region that acts as a substrate connecting a source and a drain. The biochip differentiates between ssDNA and dsDNA. (**a**) A ssDNA probe (blue) is anchored to the substrate, and a strand from the sample (red) is allowed to interact. (**b**) A ssDNA probe in suspension is allowed to interact with a strand from the sample, and forms dsDNA if complementary to subsequently interact with graphene.

**Figure 2 molecules-24-00951-f002:**
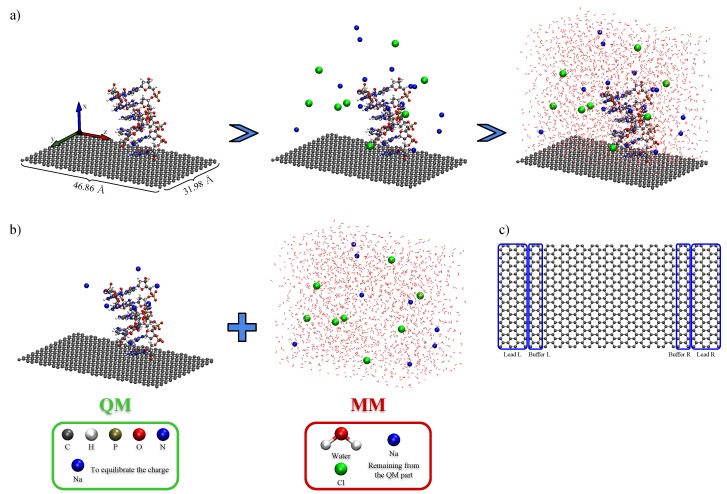
Scheme for construction (**a**) of the simulated device. The ssDNA (or dsDNA) is placed above the sheet for the vertical (or horizontal) setup, the counter-ions are added with an unbalance between positive (Na) and negative (Cl) ions and the simulation box is finally filled with water. The employed QM/MM partition (**b**) is the same for all systems: *ball-and-sticks* representation and blue spheres closest to DNA make up the QM partition and the remaining blue spheres, the green spheres and the *line* representations make up the MM partition. The electronic transport calculation is performed by dividing (**c**) the system into three regions: a central scattering region and two semi-infinite electrodes. The highlighted buffering region is used to smooth the external potential.

**Figure 3 molecules-24-00951-f003:**
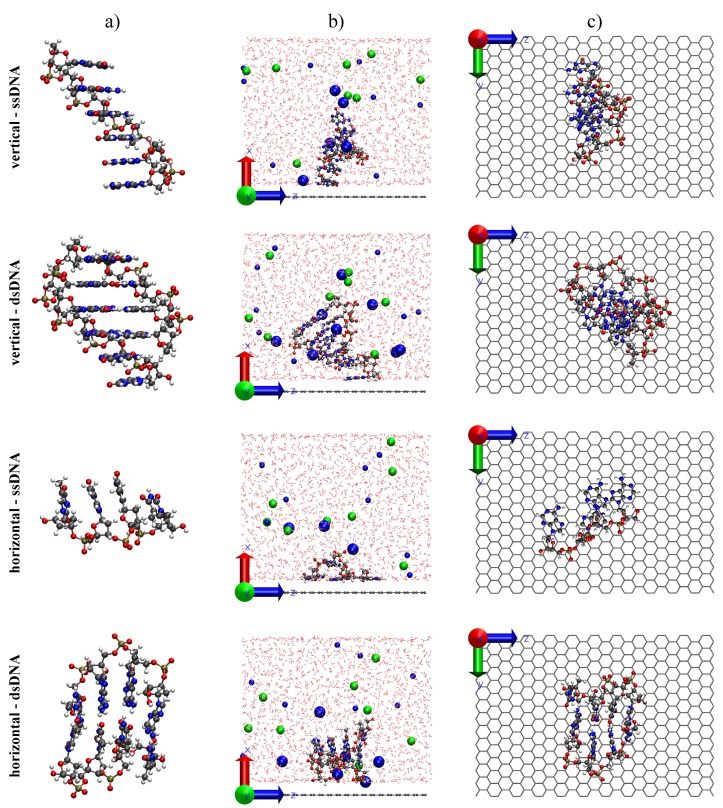
Isolated structure for ssDNA and dsDNA (**a**) in both vertical and horizontal setups for the biochip. The shown sequence is composed of six guanines (probe) for ssDNA in the vertical setup, and an addition of four cytosine for the dsDNA in the same setup, forming a partial hybrid. The two extra nucleotides of the probe are used as anchor. The sequence used for the horizontal setup was four adenine and four thymine, being the adenine sequence our ssDNA in this case. Side views (**b**) for a typical snapshot of each setup and top views (**c**) not showing the solvent for clarity.

**Figure 4 molecules-24-00951-f004:**
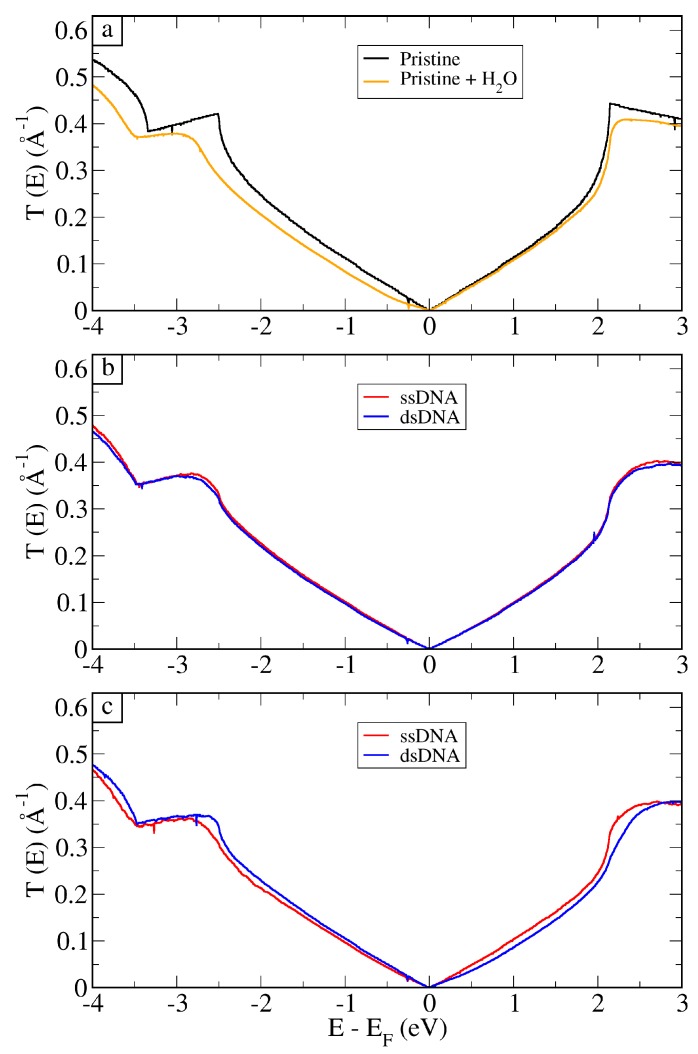
Transmission coefficient per unit width for pristine graphene and the average over 50 structures (**a**) for graphene + H${}_{2}$O. Average over 50 electronic transport calculations for ssDNA and dsDNA in the (**b**) vertical and (**c**) horizontal setups, as depicted in Figure 3.

**Figure 5 molecules-24-00951-f005:**
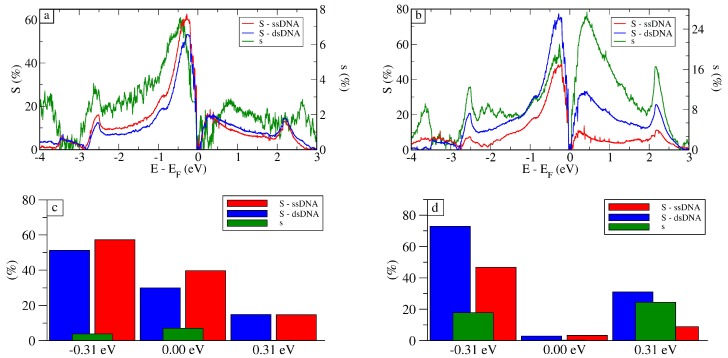
Sensitivity (S) for ssDNA and dsDNA using the results for graphene in solution as reference and selectivity (s) for dsDNA using ssDNA as reference, for the whole energy spectrum for the (**a**) vertical and (**b**) horizontal setups and for specific gate voltages for (**c**) vertical and (**d**) horizontal setups.

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
