# Peer review of "Simulating DNA Chip Design Using All-Electronic Graphene-Based Substrates"

_molecules, 2019, doi:10.3390/molecules24050951_

Reviewer 1 Report

In the paper "Simulating DNA Chip Design Using All-electronic Graphene-based Substrates", the authors theoretically studied the detection of DNA by using a graphene-based all-electronic biochip. Through the method combining quantum and classical mechanics coupled to non-equilibrium Green’s functions, the authors demonstrated the sensitivity, selectivity of the proposed device. Overall, the study can be useful for readers in this field. However, major revision should be made before this paper can be taken into further consideration. Detailed comments/suggestions are indicated as following.

1. The authors used a relatively small system to do the calculation, which cannot reflect the real situation. The DNA size is relatively large compared to the graphene sheet. The interaction of the DNA with the edge atoms could induce unwanted influence on the results. Also, the DNA may move during the MD process, and move out of the graphene edge. Could you comment on those, and add some discussions in the manuscript.

2. For better illustration, the authors are suggested to put a coordinate system to indicate the X, Y, Z directions in figure 2.

3. For figure 2, you may want to divide it into two parts, and move some of the information to the Section 3. We have to move back and forth for the current organization.

4. It’s better to have some top views that we can see the location of the DNA molecules on top of graphene.

5. In L85, the authors said “we extract 50 snapshots from the MD production run for each of the 4 setups”, what are the 4 steps? And why 50 snapshots? I couldn’t find how you used the 50 snapshots in the following discussion. If you said you averaged the data for 50 frames, there should be some error bars for both figure 3 and figure 4.

6. In Section 2 Methodology, the authors used a lot of words to describe how they built the system, which is tedious. It will be easier to understand if they can combine the description with a flow diagram.

7. The current results and discussions part includes only small part of data what the authors have obtained. They authors are suggested to add more information to support their conclusions.

8. The authors should explain each term used in equations 4 and 5.

9. The authors studied the vertical and horizontal cases, however, for real case, there must be always a combination of these two. So the authors should at least discuss how we could apply their conclusions in the real case.

10. In figure 3(c), the difference between the two curves are small. Considering about the noise induced by the system size, the choice of the K-point, the average of the data, the calculation theory, it may not be easy to do the detection based on the horizontal setup. Could you comment on that?

11. How about the influence of the NaCl concentration on the simulation data?

12. There are many DNA biosensors proposed based on either ionic current blockage or change of electronic transport properties, the author should compare the different methods, and illustrate the merits of their setup.

13. A few typos can be found in the text, please check the manuscript thoroughly.

Author Response

The answers can be found in the uploaded pdf file.

Reviewer 2 Report

Martins et. al.have used a combined quantum mechanics and molecular mechanics based simulations to study how conductivity changes in graphene surface allow detection of DNA hybridization. The study is useful for molecular detection of DNA sequences, especially those using graphene surface. They show using their simulations that signals from single stranded DNA can be distinguished from that of double stranded DNA. The results are clearly presented. I would like to see a few clarifications described below before recommending it for publication:

(1) It will be useful if the authors comment on the strength of DNA interaction in kBT with the graphene surface. For example, it will help one think if the strength of interaction strong enough to retain DNA strands when a solution flow through it.

(2) It is clear that the method can differentiate ssDNA from dsDNA. However, the authors should comment on how the identity of bases will affect detection sensitivity. This is particularly relevant since they have used rather simple DNA sequences in their analysis.

(3) I found a reference worth looking at: Detection of Nucleic Acids with Graphene Nanopores: Ab Initio Characterization of a Novel Sequencing Device. Tammie Nelson, Bo Zhang and Oleg V. Prezhdo.Nano Lett.201010 (9), pp 3237–3242. What differentiates the current work from the work presented in the above reference?

Author Response

The answers can be found in the uploaded pdf file.

Round  2

Reviewer 1 Report

I carefully read the response of the authors to the reviewers’ comments and the revised manuscript, the authors have responded to the comments regarding the scientific accuracy in a satisfactory manner. Overall, the manuscript has clearly improved and it meets now the standards to be published in Molecules. So I do recommend the manuscript to be published.